# A comparison of psychological characteristics in people with knee osteoarthritis from Japan and Australia: A cross-sectional study

Daisuke Uritani[1]*, Penny K. Campbell[2], Ben Metcalf[2], Thorlene Egerton[2]

1 Department of Physical Therapy, Faculty of Health Science, Kio University, Nara, Japan, 2 Centre for Health, Exercise and Sports Medicine, The University of Melbourne, Melbourne, Victoria, Australia

* d.uritani@kio.ac.jp

## Abstract

The aim of this study was to investigate differences in psychological characteristics between people with knee osteoarthritis (OA) from Japan and Australia. Sixty-two adults from Japan and 168 adults from Australia aged over 50 years with knee pain were included. Japanese data were collected from patients with knee OA diagnosed by medical doctors. Australian data were baseline data from a randomized controlled trial. Participants were not exercising regularly or receiving physiotherapy at the time. Psychological characteristics evaluated were depressive symptoms, fear of movement, and pain catastrophizing. These psychological characteristics were compared between the Japanese and Australian cohorts by calculating 95% confidence intervals (CIs) for difference of the mean. To test for equivalence, an equivalence margin was set at 0.5 standard deviations (SD) of the mean, where these SDs were based on the Australian data. When the 95%CI for the difference of the mean value lay entirely within the range of equivalence margin (i.e. between -0.5 and 0.5 times the Australian SD), the outcome was considered equivalent. There were no differences between the groups from Japan and Australia for depressive symptoms and the two groups were considered equivalent. There was no difference between groups for fear of movement, however the criteria for equivalence was not met. People from Japan with knee OA had higher scores for pain catastrophizing than people from Australia. The findings should be confirmed in other samples of people with knee OA from Japan and Australia due to the limitations of the participant recruitment strategy in this study. However, our findings suggest there may be a greater need to consider pain catastrophizing and build pain self-efficacy when managing Japanese people with knee OA. Implementation of international clinical practice guidelines for OA management may require different strategies in different countries due to different psychological profiles.

## Introduction

Knee osteoarthritis (OA) is a highly prevalent, chronic health condition leading to significant pain and disability among adults globally [1, 2]. In Japan, more than 60% of adults aged 60

**Data Availability Statement:** The dataset file is available from the Open Science Framework database (URL: https://osf.io/8fzs7/).

**Funding:** This study was supported by the Japanese Physical Therapy Association (H30-A11). DU received the award. The funders had no role in study design, data collection and analysis, decision to publish, or preparation of the manuscript.

**Competing interests:** The authors have declared that no competing interests exist.

years or over have radiographic knee OA and more than 26% have symptomatic knee OA [3]. However, in addition to the physical impairments, psychological impairments are common in people with knee OA, including depressive symptoms [4], fear of movement [5], and pain catastrophizing [5]. Psychological impairments in knee OA patients are associated with higher pain [4], worse physical functioning [6] and reduced physical activity levels [7, 8].

There is growing evidence to support the delivery of psychological interventions to help people with musculoskeletal pain and dysfunction including for people with knee OA [9, 10]. However, psychological interventions developed in Western countries may not be equally effective for people living in Japan. One reason for this may be differences in prevalence or severity of psychological impairments, such as depressive symptoms [4] and pain catastrophizing [11, 12]. In order to determine whether overseas evidence of psychological interventions can be applied to Japanese patients with knee OA, it is necessary to clarify whether there are differences in psychological characteristics. Thus, the purpose of this study was to compare psychological characteristics, specifically depressive symptoms, fear of movement and pain catastrophizing, between people with knee OA living in Japan and Australia.

## Materials and methods

### Study design and participants

This study was a cross-sectional study. A total of 62 people (23 males and 39 females; mean age 70.2±6.7 years) from Japan and 168 people (62 men and 106 women; mean age 62.2±7.4 years) from Australia, aged over 50 years with knee pain rated as 4 or greater on an 11-point numeric rating scale (NRS, range 0–10, higher = worse pain) were included. For the people from Japan, knee OA was diagnosed by orthopedic doctors based on radiographic images using Kellgren-Lawrence grading system (KL grade) [13]. Japanese people with KL grade ≥ 2 were included in this study. All participants were patients with knee OA undergoing conservative treatment, however they were not treated by physiotherapy. Participants were recruited at the time of seeing an orthopedic doctor from an outpatient department of a hospital. Only those who agreed to participate in the study were approached by the researchers. Japanese data were collected between December 2018 and February 2020. The data for people from Australia were the baseline data from a randomized controlled trial that evaluated the effectiveness of adding telephone coaching to a physiotherapist-delivered physical activity intervention (Australian New Zealand Clinical Trials Registry reference: ACTRN12612000308897) [14]. Participants in the Australian trial were recruited through advertisements in print, on the radio and in social media, and through a research volunteer database. They were diagnosed with knee OA using American College of Rheumatology clinical criteria [15]. They were also classified as 'sedentary' or 'insufficiently physically active' according to the Active Australia Survey [16]. Australian data were collected between July 2012 and August 2013. Exclusion criteria were the same for both groups as were: an inability to safely participate in moderate intensity exercise, undertaking regular lower extremity strengthening exercise and/or receiving physiotherapy for knee pain more than once within the past six months, knee surgery or intraarticular corticosteroid injection within the past six months, history of joint replacement on study knee or on a surgical waiting list, systemic arthritic condition, current or past (within four weeks) oral corticosteroid use, having another condition affecting lower extremity function more than knee pain, and/or if they scored more than 21 on the depression subscale of the 21-item short-form of Depression Anxiety Stress Scale (DASS-21) [17, 18].

This study was carried out in compliance with the standards laid out in the Declaration of Helsinki, and the study protocol was approved by the University of Melbourne human ethics committee (HREC no. 1137237) and the research ethics committee of Kio University (H29-

08) and Kashiba Asahigaoka Hospital (2018111002). All participants provided written informed consent to participating in the study, and for their data to be used to answer other research questions related to OA management. The trial number of this study is UMIN000027473.

## Outcome measures

Demographic details, pain, psychological characteristics (depressive symptoms, fear of movement and pain catastrophizing), and functional status were evaluated using self-report questionnaires. Participants from Japan completed questionnaires in an outpatient setting. Participants from Australia completed questionnaires at home via post/ email [14].

**Pain intensity.**   Average level of knee pain in the past week was assessed using 11-point NRS [19] with possible responses ranging from zero (no pain) to ten (worst pain possible).

**Depressive symptoms.**   Depressive symptoms were measured via the depression subscale of the DASS-21 [17, 20]. The DASS-21 [18] consists of seven items for each of the three subscales (depression, anxiety and stress). Responses range from zero (did not apply to me) to three (apply to me very much, or most of the time). Scores from each subscale are summed and multiplied by two to give a subscale score ranging from 0–42 (higher scores indicate greater level of depressive symptoms). The English version has high internal consistency and construct validity [17, 18]. A Japanese version of the DASS [17] was used for participants in Japan, however, reliability and validity of the translated version have not yet been reported.

**Fear of movement.**   Fear of movement was assessed using the Brief Fear of Movement Scale for Osteoarthritis (BFOMSO) [21]. It consists of six items extracted from the Tampa Scale for Kinesiophobia (TSK) [22] using a four-point scale from "strongly agree" to "strongly disagree" to assess fear of injury or re-injury due to movement. It ranges from six to 24 (higher scores indicate greater fear of movement). For participants in Japan, the same six questions of the BFOMSO [21] were extracted from the Japanese version of Tampa Scale for Kinesiophobia (TSK-J) [23]. The original version of TSK was translated into Japanese and linguistically validated [23]. The TSK-J is psychometrically reliable and valid for detecting fear of movement in the Japanese population suffering from neck to back pain [24].

**Pain catastrophizing.**   Pain catastrophizing was assessed using the Pain Catastrophizing Scale (PCS) [25]. It consists of 13 items, which measure tendencies to ruminate about pain, magnify pain and feel helplessness about pain, on scales from zero to four. The total score ranges from 0–52 (subscale of rumination: 0–16, magnification: 0–12, helplessness: 0–24), with higher scores indicating greater level of catastrophizing. It has high internal consistency and is associated with heightened pain, psychological distress, and physical disability among adults [26]. The Japanese version of the PCS [27] was used for the participants in Japan. The reliability and validity of the Japanese version of the PCS has been confirmed as acceptable [27]. The PCS was only collected from a total of 130 out of 168 participants in the cohort from Australia because of the high burden of the baseline questionnaire in the original study [14].

**Physical function.**   Physical function was assessed using the English and Japanese versions of the physical function subscale of the Western Ontario and McMaster Universities Osteoarthritis Index (WOMAC) Likert version [28, 29]. The physical function subscale of WOMAC has 17 questions with five response options from zero (indicating no physical disfunction) to four (indicating extreme physical dysfunction). The Japanese version of WOMAC was found to be reliable, valid, and responsive for assessing the effectiveness of total knee arthroplasty in the Japanese context despite the cultural differences from Western countries [29].

## Statistical analysis

Difference and 95% confidence intervals (CIs) between means was calculated for demographic data and outcomes including PCS subscales (rumination, magnification, and helplessness subscale). If differences in demographic data (age and Body Mass Index (BMI)) and pain between the two groups were found at a significance level of $\leq$0.05, we examined the linear relationship between those variables and psychological outcomes using Pearson's correlation coefficient to confirm whether those variables were potential covariates. If the linear correlation was r $\geq$ 0.2, the lines of best fit for the two cohorts were examined and analysis of variance used to determine if there was an interaction by group for the association. Then, if there was no interaction (no difference in the slopes), analysis of covariance controlling for the demographic variable would have been performed rather than a t-test.

To test for equivalence between means for psychological outcomes, an equivalence margin was set at 0.5 standard deviations (SD) of the mean, where these SDs were based on the Australian data. When the 95% CI of the difference of the mean value lay entirely within the range of equivalence margin (i.e. between -0.5 and 0.5 times the SD of the Australian data), we determined the outcome was equivalent. The authors started with a known sample size for the Australian participants (based on those for whom baseline RCT data [14] were available) and then calculated the sample size required for comparison. The sample size of participants from Japan required to establish equivalence of measures, with an equivalence margin of 0.5 SDs and power of 0.9, was determined to be 57. Statistical analyses were performed using SPSS software (version 26.0, SPSS Inc., Chicago, IL).

## Results

The demographic data for the two groups and main results are shown in Table 1. The groups were very similar in terms of gender, but there were significant differences between the means

**Table 1. Descriptive characteristics and results.**

| | Japan (n = 62), mean ± SD | Australia (n = 168), mean ± SD | Difference of the mean value [95%CI], (Equivalence margin minimum, maximum) |
|---|---|---|---|
| Men, n (%) | 23 (37.1%) | 62 (36.9%) | |
| Age, years | 70.2 ± 6.7 | 62.2 ± 7.4 | 7.97 [5.85, 10.09] |
| Height, cm | 159.1 ± 8.4 | 167.4 ± 9.4 | -8.25 [-10.99, -5.52] |
| Weight, kg | 65.0 ± 12.5 | 88.3 ± 20.8 | -23.31 [-27.81, -18.81] |
| BMI, kg/m$^2$ | 25.5 ± 3.9 | 31.5 ± 7.1 | -5.98 [-7.45, -4.51] |
| Pain, NRS (0–10) | 6.1 ± 1.7 | 5.7 ± 1.4 | 0.40 [-0.04, 0.84] |
| Physical function, WOMAC (0–68) | 14.6 ± 10.7 | 28.8 ± 10.7 | -14.19 [-17.32, -11.06] |
| Depression, DASS-21 (0–42) | 4.9 ± 5.4 | 4.2 ± 4.7 | 0.75 [-0.79, 2.28] (-2.35, 2.35) |
| Fear of movement, BFOMSO (6–24) | 11.7 ± 3.8 | 12.5 ± 3.2 | -0.77 [-1.77, 0.22] (-1.60, 1.60) |
| Pain catastrophizing, PCS (0–52) | 20.7 ± 11.0 | 14.8 ± 9.6[a] | 5.85 [2.79, 8.92] (-4.80, 4.80) |
| Rumination (0–16) | 8.4 ± 4.1 | 5.4 ± 3.7 [a] | 3.00 [1.83, 4.17] (-1.85, 1.85) |
| Magnification (0–12) | 3.9 ± 2.9 | 3.2 ± 2.4 [a] | 0.63 [-0.22, 1.48] (-1.20, 1.20) |
| Helplessness (0–24) | 8.4 ± 5.1 | 6.2 ± 4.2 [a] | 2.23 [0.85, 3.60] (-2.10, 2.10) |

[a]n = 130

BFOMSO: Brief Fear of Movement Scale for Osteoarthritis, BMI: Body Mass Index, DASS: Depression, Anxiety and Stress Scale, NRS: Numeric Rating Scale; PCS: Pain Catastrophizing Scale, WOMAC: Western Ontario and McMaster Universities Osteoarthritis Index.

for age and BMI, with the participants from Japan being about 8 years older and having about 6 points lower BMI.

While the Japanese group was significantly older than Australian group, there was no significant linear relationship between age and any of the psychological outcomes except for the PCS subscale of rumination (correlation r = 0.14) (Table 2). BMI was significantly correlated with the depression subscale of the DASS (r = 0.15) and the BFOMSO (r = 0.16) (Table 2). However, because these correlations did not reach our threshold (r ≥ 0.2), all psychological outcomes were compared between the two groups using t-tests.

There were no significant differences between participants from Japan and Australia for pain severity, depressive symptoms and fear of movement (Table 1). The difference in the means between Japanese and Australian cohorts for the depression subscale of the DASS and BFOMSO were within the range of the equivalence margins, however, the lower limit of the 95% CIs of differences in mean for BFOMSO (-1.77 points) was just outside the lower equivalence margin (-1.60 points), indicating some uncertainty about whether people in Japan do not have less fear of movement than people in Australia.

Participants from Japan with knee OA scored significantly higher pain catastrophizing levels than those from Australia and the difference in the means (5.85 points) was outside the margin of equivalence (-4.80 to +4.80). Looking closer at the PCS, the 'rumination' and 'helplessness' subscale scores for the participants from Japan with knee OA were significantly higher than the scores from the participants from Australia, while the subscale of 'magnification' was not different. However, the upper limit of the 95% CIs of differences in mean for PCS subscale of magnification (1.48 points) was just outside the upper equivalence margin (1.20 points), indicating some uncertainty that people in Japan do not have more tendencies to magnify about pain than people in Australia.

We performed an additional analysis to check the contribution of age and BMI as confounders in the association between country and psychological outcomes using multiple regression analysis (S1 Table). Age and/or BMI made either no contribution to the models or their contribution was to slightly strengthen the association between country and psychological outcomes.

Regarding physical function, there was a significant difference between two groups, with the participants from Japan being scored about 15 points less (i.e., better function) on the WOMAC physical function subscale (Table 1).

## Discussion

This study investigated the differences in psychological characteristics between people with knee OA living in Japan and Australia. Pain catastrophizing was higher in the participants

**Table 2. Correlation between age and BMI, and psychological outcomes.**

|  | Age | BMI |
|---|---|---|
| Depression, DASS | 0.10 | 0.15* |
| Fear of movement, BFOMSO | -0.11 | 0.16* |
| Pain catastrophizing, PCS | 0.04 | 0.01 |
| Rumination | 0.14* | -0.10 |
| Magnification | -0.10 | 0.10 |
| Helplessness | 0.03 | 0.05 |

*p<0.05; BFOMSO: Brief Fear of Movement Scale for Osteoarthritis, BMI: Body Mass Index, DASS: Depression, Anxiety and Stress Scale, PCS: Pain Catastrophizing Scale.

from Japan and the difference was accounted for by higher 'rumination' and 'helplessness' rather than by 'magnification'. On the other hand, depressive symptoms and fear of movement did not appear to be different. Of note, the two cohorts reported similar levels of pain but differed in terms of age, BMI and physical function.

The result of this study indicates that depressive symptoms among people from Japan and Australia with similar levels of pain from knee OA are equivalent. This is in contrast to a previous study that demonstrated significantly higher depressive symptoms using the Geriatric Depression Scale in community-dwelling older Japanese than Australians [30]. The previous study [30] controlled for lifestyle and health factors, while the present study did not control for any potential covariates, including lifestyle and health factors. This may be one reason for the discrepancy. Mean scores were notably low in both cohorts (4.9 and 4.2 out of 42 for participants from Japan and Australia respectively). Both Japanese and Australian cohort excluded people who scored more than 21 points on the depression subscale of the DASS-21. This meant that participants' levels of depressive symptoms were limited to the normal to moderate range [31], which may have influenced the findings.

Fear of movement did not present as being different between people with knee OA from Japan and Australia, although the lower limit of 95% CIs of mean differences in BFOMSO is slightly lower than lower equivalence margin of BFOMSO indicating uncertainty of equivalence. Given there is typically an association between pain and fear-avoidance in people with chronic pain [32], and pain levels were equivalent between our two groups, equivalence in fear of movement could be expected assuming the same association exists in both our cohorts.

Pain catastrophizing levels were significantly higher in the people from Japan, despite similar pain intensity and higher subjective physical function. The magnitude of difference in our study was relatively small compared to the clinically meaningful difference [33]. This finding is consistent with recent systematic reviews that have reported differences in pain-related beliefs and pain catastrophizing between countries, however, data was mostly from Western populations, with the exception of one Asian population (Singapore) [34, 35]. Pain catastrophizing is well known to be associated with the intensity of pain [36]. In a previous study comparing pain catastrophizing in ethnic Asian people with ethnic Westerners, Chinese-Canadians similarly reported higher pain catastrophizing scores than European-Canadians despite no difference in pain intensity from an induced painful stimulus [37].

The experience of pain is multifaceted. Sensory-discriminative aspects are those related to the location, intensity, and duration of painful stimuli, while affective-motivational aspects relate to how pain is qualitatively experienced [37, 38]. Several studies indicate that ethnic differences in pain experiences may be most apparent for the affective-motivational aspects than the sensory-discriminative aspects such as pain intensity, as affective-motivational aspects, such as catastrophizing, are more influenced by the environment and context [39–41].

The PCS subscales of rumination and helplessness appear to be higher in people with knee OA from Japan than from Australia. However, magnification did not present as being different, although with some degree of uncertainty. The subscales of rumination and helplessness are thought to become more important for longer term pain, while magnification is the predominant catastrophic cognition when pain and injury are more recent [36]. This would support our finding since knee OA is a chronic painful condition.

Other environmental and contextual factors that may explain differences in psychological characteristics include structural (e.g., healthcare systems), physical environment including climate, and culture/ethnicity differences. Several studies have shown self-efficacy, illness beliefs and the emotional and behavioral response to pain differ according to race, ethnicity and/or culture [42–44]. One theory of catastrophizing is that it is a manifestation of a broader dimension of a 'communal' approach to coping, whereby a person in pain catastrophizes in

order to garner interpersonal or social help as part of their coping strategy [36]. Catastrophizing may serve a social communicative function aimed toward maximizing the probability that distress will be managed within a social or interpersonal context rather than an individualistic context [45, 46]. Thus, greater catastrophizing may be associated with 'interdependence orientation' (emphasized in Japanese culture) rather than 'independence orientation' (predominant in Western culture) [47]. On the other hand, a recent systematic review indicated that Asian people with arthritis experience a 'lonely path' that is not shared with others to avoid being considered as complainers [48]. Japanese people with knee OA may internalize their complaints so they do not bother others [49]. These cultural differences are possible explanations for the difference in PCS, especially in helplessness, identified in our study.

Japan and Australia have broadly similar healthcare systems in terms of organization (universal healthcare with part government and part individual funding) [50], and healthcare expenditure (percentage of GDP) [51]. On the other hand, there are differences between the two countries in terms of access to healthcare. In Australia, there is good access to primary health care physicians (i.e., general practitioners) with either no or small 'out of pocket' cost. There is some access to physiotherapy through the public health system and good access to private healthcare services (without doctor referral) such as private physiotherapy where people with knee OA can get help. Meanwhile, primary healthcare by general practitioners is more limited in Japan and direct access/self-referral to physiotherapists, whether using public or private health insurance, is not permitted by law in Japan. In order to receive physiotherapy, people must see a doctor and receive a prescription for physiotherapy. These differences in type and opportunities to receive primary healthcare may contribute to the difference in pain catastrophizing if people in Japan are more worried about their knee OA because they have not seen a health professional (ruminate) and believe they will not be able to get help (helplessness).

Physical function was significantly more impaired in the participants from Australia when compared to those from Japan. This is despite pain, depressive symptoms and fear of movement being equivalent and pain catastrophizing being greater among participants from Japan. This finding was unexpected given previous studies have reported that depressive symptoms and fear-avoidance, along with pain severity, are associated with self-reported physical function [52–54]. The finding may be explained by the differences in age and/or BMI. BMI was higher in the Australian cohort and is negatively associated with self-reported function [55]. In addition, our finding is consistent with a finding that Japanese low back pain patients were significantly less impaired in functioning than American low back pain patients, despite similar pain and physical impairment findings [56]. Other studies, performed within a single (Western) population, have shown differences in associations between severity of pain and disability and people with OA from different racial and ethnic backgrounds [57, 58].

The generalizability of the results in this study is limited. These results were drawn from highly-selected samples of people with knee OA who may not be representative of broader OA populations in Japan or Australia. At inclusion, participants in this study were aged over 50 years, able to safely participate in moderate intensity exercise, were not undertaking regular exercise or receiving physiotherapy for knee pain, and were not on a waitlist for a surgical intervention. However, they had volunteered for a research project on physical activity (Australian data) or presented at a hospital for treatment of their knee pain (Japanese data). Another potential limitation is the comparability of the translation of the Japanese version of the depression subscale of DASS-21 used in the study, since validity (including cross-cultural) and reliability have not yet been reported. Differences in validity and reliability between the versions may impact on findings.

## Conclusions

People from Japan with knee OA with similar moderate to severe levels of pain had equivalent depressive symptoms and fear of movement as people from Australia, but greater pain catastrophizing. The findings need to be confirmed in other samples of people with knee OA from Japan and Australia due to the limitations of the participant recruitment strategy in this study. However, these findings suggest there may be a greater need to consider pain catastrophizing and build pain self-efficacy when managing Japanese people with knee OA. The findings also indicate implementation of international clinical practice guidelines for OA management may need different strategies for people in different countries/cultures because if differences in psychological characteristics.

## Supporting information

**S1 Table. The association of psychological outcomes with countries, age and BMI based on multiple regression analysis.**
(PDF)

## Acknowledgments

We would like to thank Professors Kim Bennell and Rana Hinman for sharing their dataset and assistance with the study methods. We would like to thank Senior lecturer Jessica Kasza for calculation of sample size, and assistance with statistical analysis and interpretation. We would also like to thank Takanari Kubo PT and Tadashi Fujii MD for recruiting Japanese participants.

## Author Contributions

**Conceptualization:** Daisuke Uritani.

**Data curation:** Daisuke Uritani, Penny K. Campbell, Ben Metcalf.

**Formal analysis:** Daisuke Uritani, Ben Metcalf, Thorlene Egerton.

**Funding acquisition:** Daisuke Uritani.

**Writing – original draft:** Daisuke Uritani.

**Writing – review & editing:** Thorlene Egerton.

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
