## [Decision Letter · Decision Letter 0]

7 Dec 2021

PONE-D-21-07731

A comparison of psychological characteristics in people with knee osteoarthritis from Japan and Australia

PLOS ONE

Dear Dr. Uritani,

Thank you for submitting your manuscript to PLOS ONE. After careful consideration, we feel that it has merit but does not fully meet PLOS ONE’s publication criteria as it currently stands. Therefore, we invite you to submit a revised version of the manuscript that addresses the points raised during the review process.

Do refer to the comments below and try to address them as closely as possible in your revised manuscript.

We look forward to receiving your revised manuscript.

Kind regards,

Maw Pin Tan, M.D.

Academic Editor

PLOS ONE

Journal Requirements:

2. Please include your ethics statement in the manuscript Methods.

3. In your Methods section, please provide additional information about the participant recruitment method and the demographic details of your participants. Please ensure you have provided sufficient details to replicate the analyses such as the recruitment date range (month and year).

4. Please ensure you have included the registration number for the clinical trial referenced in the manuscript.

Additional Editor Comments:

The two reviewers have provided rather brief comments, but Reviewer 1 would like you to revise you redid your analysis and discussion. Please do consider the reviewers' comments carefully in your revised manuscript, and provide a point-by-point rebuttal and a marked changes version of your manuscript together with your clean revised manuscript.

Reviewers' comments:

Reviewer's Responses to Questions

**Comments to the Author**

1. Is the manuscript technically sound, and do the data support the conclusions?

Reviewer #1: Partly

Reviewer #2: Yes

2. Has the statistical analysis been performed appropriately and rigorously? 

Reviewer #1: Yes

Reviewer #2: I Don't Know

3. Have the authors made all data underlying the findings in their manuscript fully available?

Reviewer #1: Yes

Reviewer #2: Yes

4. Is the manuscript presented in an intelligible fashion and written in standard English?

Reviewer #1: Yes

Reviewer #2: Yes

5. Review Comments to the Author

Reviewer #1: I thank the authors of this manuscript and the idea of analyzing important aspects of the quality of life of patients living in different countries.

Although the idea of the project sounds very interesting to me, I miss a proper discussion able to guide the reader to an understanding of the reasons behind the differences raised when comparing Japanese and Australian patients.

I suggest analyzing the Health system of the two countries and comparing the OA patients’ journey, the stereotypes associated with this disease, and possibly the different perceptions of the aging process. I think this will greatly improve the quality of the manuscript that currently seems to be more like a comparison between two different studies carried out in two different countries using different tools and it misses a discussion able to create research conclusions and questions for future studies.

Reviewer #2: This research is simply and clearly performed to help understand practitioners how to treat patients with OA in different countries.

Why did the authors do this correlational analysis (table 2)? The format of table 2 is not appropriated. The authors could look for any correlational study to see an example. Pearson's correlation is used to analyze the relation between 2 quantitative variables, and if I understand properly the aim of this study (compare Japanese and Australian samples) it does not fit because there are qualitative and quantitative variables. I cannot see where is the Student's t-test?

Spelling may be improved:

"Given the association between pain and fear-avoidance in people with chronic pain [39], equivalence in fear of movement may be expected given the equivalence in pain levels between our two groups."

On one hand [...] "On the other hand...".

6. PLOS authors have the option to publish the peer review history of their article (what does this mean?). If published, this will include your full peer review and any attached files.

Reviewer #1: No

Reviewer #2: No

---

## [Author Response · Author response to Decision Letter 0]

10 Jan 2022

Response to editor’s and reviewers’ comments

We sincerely thank you for your insightful comments. We have revised our manuscript based on the comments provided as detailed below. 

Editor

Answer: 

We corrected the title page and main text throughout the whole document based on the formatting samples. 

2. Please include your ethics statement in the manuscript Methods.

Answer: 

We have included an ethics statement in the Methods section (Lines 95-100) as follows:

- “This study was carried out in compliance with the standards laid out in the Declaration of Helsinki, and the study protocol was approved by the University of Melbourne human ethics committee (HREC no. 1137237) and the research ethics committee of Kio University (H29-08) and Kashiba Asahigaoka Hospital (2018111002). All participants provided written informed consent to participating in the study, and for their data to be used to answer other research questions related to OA management. 

3. In your Methods section, please provide additional information about the participant recruitment method and the demographic details of your participants. Please ensure you have provided sufficient details to replicate the analyses such as the recruitment date range (month and year).

Answer:

We have added additional information in the Methods section. (Lines 77, 80-83, and 85-86). Additional information included:

- “Japanese data were collected between December 2018 and February 2020”

- “Participants in the Australian trial were recruited through advertisements in print, on the radio and in social media, and through a research volunteer database.”

- Australian data were collected between July 2012 and August 2013

4. Please ensure you have included the registration number for the clinical trial referenced in the manuscript.

Answer:

We have included the registration numbers in the Methods section (Lines 80-81 and 100-101) as follows:

- “(Australian New Zealand Clinical Trials Registry reference: ACTRN12612000308897)”

- “The trial number of this study is UMIN000027473.”

‎5. ‎We note that you have indicated that data from this study are available upon request. PLOS only allows data to be available upon request if there are legal or ethical restrictions on sharing data publicly. For more information on unacceptable data access restrictions, please see http://journals.plos.org/plosone/s/data-availability#loc-unacceptable-data-access-restrictions.

Answer:

We have uploaded a dataset with on Open Science Framework 

(URL: https://osf.io/8fzs7/). 

Reviewer #1: 

1. Although the idea of the project sounds very interesting to me, I miss a proper discussion able to guide the reader to an understanding of the reasons behind the differences raised when comparing Japanese and Australian patients. I suggest analyzing the Health system of the two countries and comparing the OA patients’ journey, the stereotypes associated with this disease, and possibly the different perceptions of the aging process. I think this will greatly improve the quality of the manuscript that currently seems to be more like a comparison between two different studies carried out in two different countries using different tools and it misses a discussion able to create research conclusions and questions for future studies.

Answer:

The Discussion section has been expanded to include discussion of other possibilities to explain the differences we found and increase the interest of the article including additions to the following paragraphs:

(Lines242-244)

This finding is consistent with recent systematic reviews that have reported differences in pain-related beliefs and pain catastrophizing between countries, however, data was mostly from Western populations, with the exception of one Asian population (Singapore) [34, 35].

(Lines262-278)

Other environmental and contextual factors that may explain differences in psychological characteristics include structural (e.g., healthcare systems), physical environment including climate, and culture/ethnicity differences. Several studies have shown self-efficacy, illness beliefs and the emotional and behavioral response to pain differ according to race, ethnicity and/or culture [42-44]. One theory of catastrophizing is that it is a manifestation of a broader dimension of a ‘communal’ approach to coping, whereby a person in pain catastrophizes in order to garner interpersonal or social help as part of their coping strategy [36]. Catastrophizing may serve a social communicative function aimed toward maximizing the probability that distress will be managed within a social or interpersonal context rather than an individualistic context [45, 46]. Thus, greater catastrophizing may be associated with ‘interdependence orientation’ (emphasized in Japanese culture) rather than ‘independence orientation’ (predominant in Western culture) [47]. On the other hand, a recent systematic review indicated that Asian people with arthritis experience a ‘lonely path’ that is not shared with others to avoid being considered as complainers [48]. Japanese people with knee OA may internalize their complaints so they do not bother others [49]. These cultural differences are possible explanations for the difference in PCS, especially in helplessness, identified in our study.

(Lines279-293)

Japan and Australia have broadly similar healthcare systems in terms of organization (universal healthcare with part government and part individual funding) [50], and healthcare expenditure (percentage of GDP) [51]. On the other hand, there are differences between the two countries in terms of access to healthcare. In Australia, there is good access to primary health care physicians (i.e., general practitioners) with either no or small ‘out of pocket’ cost. There is some access to physiotherapy through the public health system and good access to private healthcare services (without doctor referral) such as private physiotherapy where people with knee OA can get help. Meanwhile, primary healthcare by general practitioners is more limited in Japan and direct access/self-referral to physiotherapists, whether using public or private health insurance, is not permitted by law in Japan. In order to receive physiotherapy, people must see a doctor and receive a prescription for physiotherapy. These differences in type and opportunities to receive primary healthcare may contribute to the difference in pain catastrophizing if people in Japan are more worried about their knee OA because they have not seen a health professional (ruminate) and believe they will not be able to get help (helplessness).

(Lines303-305)

Other studies, performed within a single (Western) population, have shown differences in associations between severity of pain and disability and people with OA from different racial and ethnic backgrounds [59, 60].

Reviewer #2: 

1. Why did the authors do this correlational analysis (table 2)? The format of table 2 is not appropriated. The authors could look for any correlational study to see an example. Pearson's correlation is used to analyze the relation between 2 quantitative variables, and if I understand properly the aim of this study (compare Japanese and Australian samples) it does not fit because there are qualitative and quantitative variables. I cannot see where is the Student's t-test?

Answer:

In the statistical analysis section, if differences in demographic data (age and Body Mass Index (BMI)) and pain between the two groups were found at a significance level of ≤0.05, we examined the linear relationship between those variables and psychological outcomes using Pearson’s correlation coefficient to confirm whether those variables were potential covariates. If the linear correlation was r ≥ 0.2, the lines of best fit for the two cohorts were examined and analysis of variance used to determine if there was an interaction by group for the association. Then if there was no interaction (no difference in the slopes), analysis of covariance controlling for the demographic variable would have been performed rather than a t-test. We found there were no linear relationships reaching our threshold (r ≥ 0.2) between age and BMI, and any of the psychological outcomes in table 2. Comparisons of demographic data and psychological outcomes using t-tests are shown in table 1. 

2. Spelling may be improved:

"Given the association between pain and fear-avoidance in people with chronic pain [39], equivalence in fear of movement may be expected given the equivalence in pain levels between our two groups."

On one hand [...] "On the other hand...".

Answer:

These sentences have been edited and now read (Lines235-238) :

- “Given there is typically an association between pain and fear-avoidance in people with chronic pain [32], and pain levels were equivalent between our two groups, equivalence in fear of movement could be expected assuming the same association exists in both our cohorts.” 

- “On the other hand…” has been changed to “However,…”

---

## [Decision Letter · Decision Letter 1]

28 Feb 2022

PONE-D-21-07731R1A comparison of psychological characteristics in people with knee osteoarthritis from Japan and Australia: A cross-sectional studyPLOS ONE

Dear Dr. Uritani,

Thank you for submitting your manuscript to PLOS ONE. After careful consideration, we feel that it has merit but does not fully meet PLOS ONE’s publication criteria as it currently stands. Therefore, we invite you to submit a revised version of the manuscript that addresses the points raised during the review process.

The reviewers have indicated that your manuscript should be accepted pending some minor changes. Please do address the changes accordingly and resubmit at your soonest convenience.

We look forward to receiving your revised manuscript.

Kind regards,

Maw Pin Tan, M.D.

Academic Editor

PLOS ONE

Journal Requirements:

Reviewers' comments:

Reviewer's Responses to Questions

**Comments to the Author**

1. If the authors have adequately addressed your comments raised in a previous round of review and you feel that this manuscript is now acceptable for publication, you may indicate that here to bypass the “Comments to the Author” section, enter your conflict of interest statement in the “Confidential to Editor” section, and submit your "Accept" recommendation.

Reviewer #2: All comments have been addressed

Reviewer #3: All comments have been addressed

2. Is the manuscript technically sound, and do the data support the conclusions?

Reviewer #2: Yes

Reviewer #3: Partly

3. Has the statistical analysis been performed appropriately and rigorously? 

Reviewer #2: Yes

Reviewer #3: No

4. Have the authors made all data underlying the findings in their manuscript fully available?

Reviewer #2: Yes

Reviewer #3: Yes

5. Is the manuscript presented in an intelligible fashion and written in standard English?

Reviewer #2: Yes

Reviewer #3: Yes

6. Review Comments to the Author

Reviewer #2: In my opinion, authors have improve the manuscript properly.

Author did explain the reasons why they use correlational analysis.

Spelling has been changed and improved.

Reviewer #3: This is interesting study comparing psychological aspect of people suffering with Osteoarthritis from 2 developed countries; Australia and Japan.

My only concern is the current data analysis is not sufficient to draw the conclusion as suggested. I would suggest the authors to run linear regression to look at whether the higher Pain catastrophizing rumination reported by Japanese as compared to Australia is influence/confound by other factors such age and BMI.

Example of Table is as attached in reviewer attachment

7. PLOS authors have the option to publish the peer review history of their article (what does this mean?). If published, this will include your full peer review and any attached files.

Reviewer #2: No

Reviewer #3: **Yes: **Sumaiyah Mat

---

## [Author Response · Author response to Decision Letter 1]

9 Apr 2022

Response to reviewers

My only concern is the current data analysis is not sufficient to draw the conclusion as suggested. I would suggest the authors to run linear regression to look at whether the higher Pain catastrophizing rumination reported by Japanese as compared to Australia is influence/confound by other factors such age and BMI.

Answer 

We performed multiple regression analysis as suggested (S1 Table). In most models, age and BMI had no significant associations with psychological outcomes, or even if there were significant associations, they were extremely small. These results do not show any inconsistency with the results shown in Table 2.

We added the following sentences from line 166 to 170 and S1 Table.

“We performed an additional analysis to check the contribution of age and BMI as confounders in the association between country and psychological outcomes using multiple regression analysis (S1 Table). Age and/or BMI made either no contribution to the models or their contribution was to slightly strengthen the association between country and psychological outcomes.”

---

## [Editor Report · Decision Letter 2]

19 Apr 2022

A comparison of psychological characteristics in people with knee osteoarthritis from Japan and Australia: A cross-sectional study

PONE-D-21-07731R2

Dear Dr. Uritani,

We’re pleased to inform you that your manuscript has been judged scientifically suitable for publication and will be formally accepted for publication once it meets all outstanding technical requirements.

Kind regards,

Maw Pin Tan, M.D.

Academic Editor

PLOS ONE
---

## [Editor Report · Acceptance letter]

26 Apr 2022

PONE-D-21-07731R2 

A comparison of psychological characteristics in people with knee osteoarthritis from Japan and Australia: A cross-sectional study 

Dear Dr. Uritani:

I'm pleased to inform you that your manuscript has been deemed suitable for publication in PLOS ONE. Congratulations! Your manuscript is now with our production department. 

Kind regards, 

on behalf of

Dr. Maw Pin Tan 

Academic Editor

PLOS ONE